# MG53, A Tissue Repair Protein with Broad Applications in Regenerative Medicine

**DOI:** 10.3390/cells10010122

**Published:** 2021-01-11

**Authors:** Zhongguang Li, Liyang Wang, Huimin Yue, Bryan A. Whitson, Erin Haggard, Xuehong Xu, Jianjie Ma

**Affiliations:** 1Department of Surgery, The Ohio State University Wexner Medical Center, Columbus, OH 43210, USA; zhongguanglee@snnu.edu.cn (Z.L.); Bryan.Whitson@osumc.edu (B.A.W.); Erin.Haggard@osumc.edu (E.H.); 2Laboratory of Cell Genetics and Developmental Biology, Shaanxi Normal University College of Life Sciences, Xi’an 710062, China; wly1826@snnu.edu.cn (L.W.); yuehuimin@snnu.edu.cn (H.Y.)

**Keywords:** TRIM protein, cell membrane repair, muscular dystrophy, myocardial infarction, acute kidney injury, acute lung injury

## Abstract

Under natural conditions, injured cells can be repaired rapidly through inherent biological processes. However, in the case of diabetes, cardiovascular disease, muscular dystrophy, and other degenerative conditions, the natural repair process is impaired. Repair of injury to the cell membrane is an important aspect of physiology. Inadequate membrane repair function is implicated in the pathophysiology of many human disorders. Recent studies show that Mitsugumin 53 (MG53), a TRIM family protein, plays a key role in repairing cell membrane damage and facilitating tissue regeneration. Clarifying the role of MG53 and its molecular mechanism are important for the application of MG53 in regenerative medicine. In this review, we analyze current research dissecting MG53′s function in cell membrane repair and tissue regeneration, and highlight the development of recombinant human MG53 protein as a potential therapeutic agent to repair multiple-organ injuries.

## 1. Introduction

The cell membrane allows a complex communication and exchange between the inside of a cell and its extracellular environment. Within a life time, cells may be injured by different factors including mechanical stress, radiation such as UV light, and biochemical drugs. If the membrane injury is not repaired in time, the injury will progress to cell death and permanent tissue damage. It is critical to maintain cellular integrity to ensure cell survival and tissue regeneration, as defects in cell membrane repair are linked to the pathophysiology of many human diseases including muscular dystrophy, heart failure, lung injury, and kidney disease. As a rapidly developing field, regenerative medicine aims to repair or replace damaged cells, tissues, and organs [1] A potential therapeutic approach targeting cell membrane repair for regenerative medicine is the recently discovered Mitsugumin 53 (MG53).

MG53 is a tripartite motif-containing (TRIM) family protein and plays a key role in repairing cell membrane damage and facilitating tissue regeneration. MG53 was first identified from skeletal muscle using a novel immunoproteomic approach described by Weisleder, Takeshima, and Ma [2]. In 2009, we reported that MG53 acted as a key component of the cell membrane repair machinery, and demonstrated that mice with ablation of MG53 (*mg53^−/−^*) display compromised sarcolemma repair with progressive myopathy [3]. MG53 initiates the assembly of repair-patch formation through facilitating the movement of intracellular vesicles to the membrane injury site. In heart studies, *mg53^−/−^* mice have shown increased susceptibility to stress-induced myocardial infarction due to impaired cardiomyocyte repair function [4]. Subsequent studies revealed pathology in the kidney, lung, and cornea of *mg53^−/−^* mice, which are also linked to defective cell membrane repair function of the affected tissues [5,6,7]. The therapeutic benefits of the recombinant human MG53 (rhMG53) protein in treatment of stress-induced injuries to the skeletal muscle, heart, lung, kidney, cornea, brain, liver, and skin were established in different animal models [5,6,7,8,9,10,11,12,13,14,15,16,17,18,19,20,21] (Table 1). 

## 2. Diverse Distribution of MG53 and Cellular Mechanism of MG53-Mediated Regeneration

MG53 is predominantly expressed in striated muscles [3]. Studies show that low levels of MG53 are also present in the lung epithelial cells, inner cortex of the kidney, along within the tear film, corneal epithelia, and aqueous humor [5,6,7,13,14,18]. MG53 can be secreted from skeletal muscle and circulates throughout the entire body to reach all tissues and organs [22,23,24]. This protein is highly conserved in many other species in the animal kingdom, which preserves its fundamental and universal biology functions [3]. 

### 2.1. MG53 Protein Structure, Distribution, and Conservation across Species

More than 80 known TRIM protein genes have been identified in humans so far, which all share the RING B-box coiled-coil (RBCC) motif. The RBCC domain comprises three motifs which are a RING-finger domain, one or two B-box domains, and a coiled-coil domain. Their cellular functions are diverse including cell proliferation, differentiation, development, oncogenesis, apoptosis, protein quality control, autophagy, innate immunity, and retroviral replication [25,26,27,28,29,30]. Genomic analysis of the TRIM family reveals that the human TRIM family is split into two groups that differ in domain structure, genomic organization, and evolutionary properties. MG53 belongs to group 2, which is characterized by the presence of a carboxyl-terminal SPRY domain, a repeated sequence in the dual-specificity kinase SplA, Ca^2+^-release channel ryanodine receptors (RyR), and a unique set of genes in each mammal examined [31].

Molecular phylogenetic analyses reveal that MG53 forms a close association among other organisms (Figure 1A). The human MG53 shares homology with lots of mammalian, especially primates including orangutan/*Pongo abelii* (99.6%), gorila/*Troglodytes gorilla* (99.4%) and Tcardiol/*Hylobates moloch* (99.2%). The homology indicates that diverse animals with the MG53 gene have a significant close evolutionary relationship that might have originated from a common ancestor. 

Based on the phylogenic tree, MG53 proteins are divided into two categories according to the variants of amino acids in different domains, such as RING finger, B-box zinc, and SPRY domain (Figure 1B and Table 2). Based on their functional domains, MG53 homology can be grouped into two categories. The first category, represented by human MG53 protein, consists principally of three functional domains: RING finger, B-box zinc, and SPRY domain. Members of the second category, including mouse and rat, contain additional domains such as Poly (hydroxyalcanoate) granule-associated protein (phasin) and SPRY-associated domain (PRY). The RING finger domain mediating ubiquitination is the characteristic signature of E3 ubiquitin ligase and the zinc-finger motif of the MG53 through binding two zinc cations to Cys3HisCys4 amino acid. However, the specific function of the B-box domain remains unclear. In general, the PRY-SPRY domain is recognized as more evolutionarily ancient, which conveys the selectivity and specificity of its E3 activity [32,33]. Mutations found in the SPRY-containing proteins can cause Mediterranean fever and Opitz syndrome [32,33].

MG53 is highly conserved among different species, including human, monkey, rat, and mouse [3,32,33] The RING finger motif that contains a Cys3HisCys4 amino acid motif is a zinc-finger motif binding with two Zn^2+^ [3]. The B-box domain is another zinc-binding motif, and its specific function may be linked to cell membrane repair and wound healing [36,37]. Coiled-coil domains mediate homo- or hetero-oligomerization of TRIM family proteins, especially for their self-associations and interactions with different binding partners [37]. The PRY-SPRY domain is believed to be a central mediator for selective interaction with its partners [38]. A critical cystidine residue (C242) in the PRY-SPRY domain is critical for MG53 oligomer formation to facilitate dysferlin or annexin fusion vesicles at the membrane injury sites [3,36,37].

MG53 is abundantly distributed in mouse and rat skeletal muscle and heart, but its expression level is very low in hearts of human, porcine, and [33] species along with no expression in liver, skin, and brain [3,19,21]. MG53 also locates in the kidney and lung with much lower expression than that in heart and skeletal muscle [5,6,13,14]. Native MG53 distributed in the corneal epithelia, tear film, and aqueous humor suggests its potential function in corneal homeostasis [7,24]. Interestingly, MG53 can be secreted from the muscle into the blood circulation and participates in multiple physiologic and pathologic processes, in particular for membrane repair in non-muscle organs [7,19,20,21].

To expand the mechanistic evaluation of MG53 in cell membrane repair and tissue regeneration, we have created transgenic mice with constitutive secretion of MG53 in the blood stream (tPA-MG53) [9]. The tPA-MG53 mice lived a healthy life-span. These mice exhibited significant enhanced capacity of tissue regeneration without alteration of glucose handling in metabolism [9].

### 2.2. MG53 and Its Molecular Mechanism in Cell Membrane Repair

Striated muscle cells undergo severe membrane stress in response to muscle contraction, so acute muscle membrane repair is particularly important [39]. Several molecular components such as dysferlin and CaV3 were identified in its repair, particularly those specific to cardiac and skeletal muscles [40,41,42,43,44]. Cai et al. first reported that MG53 was a central component of the plasma membrane repair machinery and facilitates repair of acute membrane damage in an oxidation-dependent manner [3,45,46]. Genetic ablation of MG53 in mice (*mg53^−/−^*) resulted in defective membrane repair function in striated muscle and led to progressive skeletal myopathy [3].

The function for MG53 to repair acute cell membrane injury is illustrated in Figure 2, where red fluorescent protein (RFP)-labeled MG53 was transiently expressed in C2C12 cells. Prior to injury and at resting state, RFP-MG53 was present on the intracellular vesicles, plasma membrane, and in the cytosol. Poking of the cell by a microelectrode resulted in rapid translocation of RFP-MG53 containing vesicles to the acute injury site. Injury of the cell led to transient change of the redox state from reduced environment to oxidized environment, which triggered redox-dependent oligomerization of MG53 allowing for formation of the membrane repair patch [3,47].

The MG53-mediated membrane repair process is regulated by multiple factors. Zn^2+^ can interact with MG53 to promote cell membrane repair because both RING and B-box motifs of MG53 have Zn^2+^-binding domains which contribute to MG53-mediated repair [36]. Leucine zipper motifs in the coiled-coil domain of MG53 enrich understanding of the mechanism that facilitates oligomerization of MG53 during membrane repair. Both oxidation of the thiol group of Cys242 and leucine zipper-mediated interaction among the MG53 molecules contribute to the nucleation process for MG53-mediated membrane repair [37]. Extracellular Ca^2+^ facilitates repair vesicles’ fusion to reseal the membrane [3].

Vesicle-related proteins are involved in the regulation of the MG53-mediated membrane repair process. Polymerase I and transcript release factor (PTRF) acts as a docking protein leading MG53 to the membrane during the repair via possible binding to exposed membrane cholesterol at the injury site. Specific RNA silencing of PTRF leads to defective muscle membrane repair, and overexpression of PTRF can rescue membrane repair defects in dystrophic muscle [48]. Furthermore, as a key cytoskeleton motor protein, non-muscle myosin type IIA (NM-IIA) interacting with MG53 essentially regulates vesicle trafficking [49], along with interactions of MG53 with both dysferlin and CaV3 during cell membrane repair [8,50]. Dysferlin and annexin A1, along with MG53, play direct roles at sites of damaged membrane for remodeling of the transverse tubule system, whereas CaV3 indirectly affects membrane resealing because of altered trafficking [51].

Some amino acids on the MG53 primary sequence are required for MG53-mediated membrane repair. A single amino acid replacement, K279A, leads to severe aggregation of MG53 within inclusion bodies in HeLa cells. Due to the loss of the positive charge, the localization and function of dysferlin and MG53 are significantly changed [52]. Recently, MG53-associated membrane repair was linked to a mono-ADP-ribosylation in wound healing following myocardial injury. ADP-ribosylation interfered with assembly of MG53 repair complexes in MG53 R207K and R260K mutations [53]. As the target of S-nitrosylation (SNO) and oxidation SNO, with C144 mutation, MG53 C144S prevented oxidation from oxidation-induced degradation of MG53, which resulted in its preservation of the protein and enhanced cell survival following oxidative insult [54,55].

## 3. MG53 in Cardiac Protection against Injury from Ischemia/Reperfusion (I/R)

Blockage of heart blood flow leads to myocardial ischemia. Persistent ischemia causes myocardial infarctions, resulting in profound myocyte death, irreversible myocardial damage, and a permanent loss of contractile mass. The death of cardiomyocytes cannot be supplemented by newly generated cells. Reperfusion can trigger further damage to the myocardium; i.e., I/R injury via reactive oxygen species-induced oxidative stress, calcium overload, or calpain activation [56,57,58].

Cao et al. reported that MG53 is downregulated in cardiac I/R injury and increase of MG53 expression appears cardioprotective on I/R-, hypoxia-, or oxidative stress-induced myocardial damage [11]. *mg53^−/−^* mice suffered exaggerated I/R injury, but over-expression of MG53 was sufficient to protect cardiomyocytes against hypoxia- and oxidative stress-induced cell death. The interaction between MG53 and CaV3 can activate pro-survival kinases and further promote cardioprotection [11]. The genetic ablation of *mg53* increases susceptibility to I/R-induced myocardial damage with decelerated membrane resealing compared with wild-type controls [4,16]. Routinely, stress conditions such as myocardial I/R injury can lead to loss of mitochondrial function and subsequent irreplaceable loss of cardiomyocytes. Wang et al. proved that the cardiac protection of the MG53-mediated membrane repair is cholesterol-dependent against these stress conditions with important therapeutic implications [16]. MG53 and the CaV3 complex is required for the p85 subunit of phosphoinositide 3-kinase (p85-PI3K) via PostC-mediated activation of the reperfusion injury salvage kinase (RISK) pathway [4].

In δ-SG-deficient TO-2 hamsters with severe cardiac dysfunction and dilated congestive heart failure, muscle-specific overexpression of human MG53 gene via adeno-associated virus (AAV), and systemic delivery of rhMG53, enhanced membrane repair in the cardiomyocytes via ameliorated pathology with significant improvement of heart functions in the genetically modified hamsters [17]. The overexpressed MG53 increased the dysferlin expression and facilitated its trafficking to damaged muscle membrane. The cardiac function improvement promotes phosphorylation ERK1/2, Akt, and glycogen synthase kinase (GSK) expression, and reduces the pro-apoptotic genes’ expression in the MG53 cardiomyocytes [17]. The rhMG53, introduced into the injured cardiomyocytes via perfusion, directly triggers activation of the PI3K survival pathway for cardiac protection in pigs [12].

S-nitrosylation can reduce IR-induced MG53 degradation along with infarct size in a perfused heart [54,55] and the mono-ADP-ribosylation cycle is essential for oligomerization of MG53 at the I/R injury site [53]. While the ADP-ribosylated MG53 would be considered as a rapid and abundant biomarker of cardiac ischemic injury, MG53 not only facilitates plasma membrane repair, but also further activates the intracellular RISK pathway for cardioprotection. Although cardioprotective roles for endogenous myocardial MG53 could not be extrapolated from rodents to humans, potential therapeutic application of rhMG53 for myocardial membrane injury prevails [59].

## 4. MG53’s Role in Repair and Regeneration of Skeletal Muscle

Within the routine contraction and relaxation of skeletal muscle, sarcolemma injury occurs under physiological conditions in an average life span. Impaired membrane repair capacity has been linked to many different disease states, such as muscular dystrophy. Cai et al. reported *mg53^−/−^* mice showed progressive myopathy and reduced exercise capability associated with defective membrane-repair capacity [3]. For the tPA-MG53 mice, exercise-induced muscle injury was less than that in the controls, and tPA-MG53 muscles recovered faster in the presence of sustained elevation of MG53 in blood circulation [9]. MG53 significantly improved muscle satellite cell (mSC) proliferation and increased regenerative capacity related to muscle injury [9].

Another protein related to skeletal muscle membrane repair, dysferlin, is recognized as a fusion protein involved in restoration of muscle membrane integrity after muscle injury [60,61,62]. MG53-mediated active trafficking of intracellular vesicles to sarcolemma involves the movement of dysferlin to injured sites on cell membranes during repair patch formation [8]. Another repair-related protein, Cav3, is critical to damage recovery in cell membranes repaired by MG53. P104L and R26Q mutations of Cav3 resulted in their remaining in the Golgi apparatus. This defect resulted in the dis-localization of both MG53 and dysferlin, leading to defects in the membrane repair. These data indicate that dysferlin, Cav3, and MG53 are essential in repairing damage to muscle membrane by forming a molecular complex in muscle and heart [8].

Dystrophic muscle pathology is mainly thought to involve δ-sarcoglycan (δ-SG). TO-2 hamsters with δ-SG deletion develop chronic damage of muscle fibers along with leakage of sarcolemma [63]. He et al. demonstrated that muscle-specific overexpression of human MG53 enhanced the membrane repair of δ-SG deficient TO-2 hamsters with significantly improved pathological conditions and amended muscle function [17]. Application of rhMG53 through multiple approaches including intramuscular injection (IM), intravenous injection (IV), and subcutaneous injection (SC) can all improve the membrane repair capacity of skeletal muscle fibers, as well as ameliorate some of the pathology associated with muscular dystrophy in animal studies with dystrophin-deficient *mdx* mouse [10].

rhMG53 applied via IV to the wild-type mice protected I/R injured muscle and improved pathologic syndrome of skeletal muscle dystrophy [64]. However, rhMG53 did not protect against I/R injury in rat skeletal muscle. This was likely due to the fact that the serum concentrations of MG53 in mouse and human are significantly lower than that in rat, which makes rat muscle less sensitive to the therapeutic effects of rhMG53 [64].

Burn injury induced a severe failure of plasma membrane repair, resulting in skeletal muscle membrane injury. Burn-induced inactivation of disulfide isomerase can inhibit MG53 dimerization and result in failure of musculoskeletal membrane repair [65]. Gushchina et al. proved that short-term rhMG53 treatment is able to increase sarcolemma integrity, independent of the canonical dysferlin-mediated and Ca^2+^-associated pathway, which is important for sarcolemmal membrane repair in the dysferlin-deficient mouse model. rhMG53 improves muscle fiber survival, which could be beneficial for patients with type 2B limb girdle muscular dystrophy (LGMD2B) [66].

As a small molecule native to striated muscle, MG53 can be released into the extracellular space to directly reboot resealing capacity of local membrane of the tissue. Thus, circulating MG53 might play a role in muscle injury-induced regeneration. Targeting the extracellular MG53 function could promote beneficial effects on human diseases associated with defective membrane repair capacity within the entire body. The above findings suggest that MG53 could be translated into a treatment for use in human patients to target cell membrane repair in regenerative medicine.

## 5. Injury Protection of MG53 in Non-Muscle Organs

In addition to MG53′s protective role in cardiac and muscular pathology, MG53-mediated repair machinery protects against injury to non-muscle organs including I/R or contrast-induced acute kidney injury [5,15], stress-induced damage in lungs [6,13,14,67], chronic skin wounds [68], and hepatic I/R injury in liver transplantation and hepatic resections [20]. The most recent publication on the function of MG53 predicts more involvement of the protein in other diseases [69].

### 5.1. MG53 on Acute Kidney Injury (AKI)

Although the native MG53 protein is mainly expressed in striated muscle, it is also present in the kidney at low but significant concentrations, especially in proximal tubular epithelium (PTE) cells [5]. PTE cells are extremely susceptible to membrane damage under stress conditions such as chemotherapy, I/R, nephrotoxin, or sepsis [5,15]. *mg53^−/−^* mice developed renal pathological phenotypes and renal tubulointerstitial injury without affecting glomeruli. Compared with wild-type control, *mg53^−/−^* mice have increased sensitivity to I/R-induced acute kidney injury (AKI). rhMG53 administration to injury sites facilitated repair after I/R injury without adverse effects [5].

Moreover, IV delivery of rhMG53 ameliorated cisplatin-induced AKI without affecting its anti-tumor function [5]. Liu et al. reported that MG53 protected against contrast-induced AKI (CI-AKI) by reducing cell membrane damage and apoptosis [15]. Using an animal scalding model of 30% of total body surface area (TBSA), another group investigated the role of MG53 in the protection of kidneys after severe burn injury. IV injection of rhMG53 reduces the mortality and the histological alternation of renal tubular epithelial cells after burn damage [18]. These data present evidence that MG53 is a vital component of reno-protection, and targeting of MG53-mediated repair of PTE cells has potential to both prevent and treat AKI and nephrotoxin exposure.

### 5.2. MG53-Mediated Repair in Wound Healing

Li et al. developed a bioinspired hydrogel to deliver rhMG53 for chronic wound healing [68]. This approach facilitates rapid delivery of rhMG53 to improve the epithelium recovery of wounds along with continual release of the protein for persistent treatment of the wound [21,68]. Accordingly, *mg53^−/−^* mice exhibited remarkable defects in skin architecture and elevated collagen deposition in the dermal epithelium. Without MG53, the mice displayed abnormal scarring in their healing with poor regenerative capacity. After rhMG53 administration, wound re-epithelialization was promoted with reduced post-injury fibrosis and vascularization via promotion of membrane repair [21]. The MG53-mediated re-epithelialization on injured skin accomplishes the repair via interfering with TGF-β-dependent activation of myofibroblast differentiation for reduced scarring [21].

### 5.3. MG53-Mediated Repair in Liver Protection

Hepatic ischemia-reperfusion (I/R) injury mainly occurs during liver transplantation and hepatic resection [70]. Hepatic I/R induces severe damage to cell membrane hepatocytes (also known as epithelial cells in liver), which leads to hepatocyte death and the subsequent hepatic I/R injury [71]. Yao et al. proved that rhMG53 administration reduced the release of LDH, AST, and ALT post-hepatic I/R injury, and decreased liver oxidative stress. Subsequently, rhMG53 treatment enhanced dysferlin expression which is co-localized with MG53 in rat hepatic I/R injury models [20]. Furthermore, the study showed that down-regulation of dysferlin can reverse the protective effects of MG53 on damaged hepatocyte I/R [20]. Dysferlin-anchored MG53 can reduce the oxidative stress and cell apoptosis for MG53-mediated hepatic damage repair [20].

### 5.4. MG53-Mediated Repair in Cornea Protection

Corneal wound healing is a complex and coordinated process, involving repair to the epithelial layer, migration of viable epithelial cells and fibroblasts for wound closure, and stimulation of cellular proliferation for tissue regeneration. Prevention of excessive stromal myofibroblast activation and vascular in-growth is also imperative to avoid fibrosis and angiogenesis, which can compromise the transparency of the cornea. An approach that can functionally target multiple steps in corneal wound healing may have the potential to significantly improve healing outcomes, leading to novel therapeutic options.

We found that native MG53 is present in the corneal epithelia, tear film, and aqueous humor, suggesting its potential function in corneal homeostasis [7]. *Mg53^−/−^* mice display impaired healing and regenerative capacity following alkaline injury to the cornea, and rhMG53 promotes re-epithelialization and reduces post-injury fibrosis and vascularization. Exogenous rhMG53 protein protects the corneal epithelia against mechanical injury. We also demonstrated that rhMG53 modulates TGF-β-mediated fibrotic remodeling associated with corneal injury [7]. These findings support the bi-functional role of MG53 in facilitating corneal healing and maintaining corneal transparency by reducing fibrosis and vascularization associated with corneal injuries.

## 6. A Potential Role of MG53 in Treatment of Acute Lung Injury

Various pathological stresses including sepsis, trauma pneumonia, and blood transfusion can cause acute lung injury (ALI) [72]. We have shown that *mg53^−/−^* mice display abnormal lung structure and function under basal conditions, a finding consistent with the observation that MG53 is present in the alveolar epithelia, where ablation of the MG53 gene leads to defective alveolar structure in the mutant mice [6]. Upon ventilation-induced lung injury, *mg53^−/−^* mice show exacerbated damage responses compared to wild-type littermates [6].

rhMG53, when administered either intravenously or via aerosolization, has the ability to effectively mitigate lung ischemia-reperfusion injury, lipopolysaccharide (LPS) induced inflammation, and porcine pancreatic elastase (PPE)-induced emphysema in rodents. rhMG53 protects against stress-induced injury to both lung epithelial and endothelial cells. Repetitive administration of rhMG53 improves pulmonary structure associated with chronic lung injury in mice [6]. Published data from other investigators also confirm the beneficial effects of inhalational rhMG53 to treat lung injury [13,67].

As one of the world’s greatest public health challenges, influenza causes 1 billion cases resulting in 290,000 to 650,000 influenza-related deaths globally every year [73]. Recently, the therapeutic potential activity of MG53 coupling its anti-inflammasome properties with lung regeneration has not only provided a promising approach to develop MG53 to combat the lethal influenza virus H1N1, but also a strong potential treatment for the lethal Covid-19 SARS-CoV-2 virus [74].

Several other TRIM proteins have antiviral effects [75,76,77,78]. However, Sermersheim et al. have demonstrated that MG53 does not have an impact on influenza viral titers; however, it does have a protective impact through mitigation of pyroptosis [74]. By knockdown inhibition of MG53 in cultured THP-1 cells, they demonstrated that macrophages deficient in MG53 secreted increased levels of interferon-β and interleukin-1β. The MG53 knockdown cell displayed nearly a twofold increase in phosphorylation of p65 at serine 536, a well-known NF-κB activation marker. Immunofluorescent imaging revealed that along with elevated p-p65, MG53 suppression correspondingly resulted in more p65 localization to the nucleus. These observations lead one to conclude that MG53 serves as a negative regulator of NF-kB and the potential mechanism of modulation of inflammatory mediators, which are critical to an infectious response [69].

Kenney and Li et al. [74] found that exogenous injection of rhMG53 through the tail vein protected mice from the lethal influenza virus strain A/PR/8/34 (H1N1) infection. Twenty-four hours after being infected with influenza virus, mice were injected with rhMG53 (2 mg/kg) or saline via tail vein as a control daily for up to 14 days. All control group mice injected with saline died within 9 days post-infection. In sharp contrast, the mice treated with rhMG53 had a survival rate of 92% and recovered to normal body weight by day 14 post-infection [74]. A separate cohort of infected and treated mice was sacrificed 7 days post-infection to assess the protective role of rhMG53 during virus infection. Histological evaluation at 7 days post-infection demonstrated that the alveolar structure of the rhMG53 treatment group was significantly preserved along with lung injury score decrease. The infiltrating CD11b positive lymphocytes were significantly reduced in the rhMG53 treatment group. ELISA quantification showed that inflammatory cytokines of IFNβ, IL-6, and IL-1β were significantly lower in lung tissues derived from rhMG53-treated mice compared with saline control mice. Protein expression analysis of lung tissue showed that the NLRP3 was significantly increased after viral infection, but was significantly reduced by rhMG53 treatment subsequently along with significant reduction of the key protein of cell pyroptosis [79], GSDMD-N terminal, after rhMG53 treatment.

rhMG53 develops a promising host-directed therapeutic for infectious lung injury by suppressing cytokine storm and virus-induced tissue damage [74]. It seems that the current life-threatening COVID-19 pandemic shares the same immunoreaction mechanism with that of the lethal influenza H1N1 virus, which exposes a unique opportunity to design MG53-associated treatment to heal lung damage resulting from the COVID-19 pandemic.

## 7. Alteration of MG53 Expression Does Not Impact Glucose Metabolism

As the world knows, diabetes mellitus (DM) is a systemic disease that has enormous impact on multiple organ systems, quality of life, and economics. It has been proposed that modulation of circulating MG53 levels through administration of monoclonal antibodies may have a role in treating DM [22]. While it is worrisome that elevated MG53 levels may contribute to glucose insensitivity, the approach to assessment of the antibody levels necessitates more in-depth analysis and refinement of assessment modalities [80].

Wang and colleagues worked to understand any contribution of MG53 in development of DM [24]. In their elegant research published in *Diabetes* [24], they were able to demonstrate a null impact of whole-body ablation of MG53 or sustained elevation on glucose handling and insulin signaling. In that work, Wang et al. created diabetic MG53 knockout mice (db/db-mg53*^−/−^*) and diabetic mice with constitutively overexpressed MG53 (db/db-tPA-MG53) though murine husbandry by crossbreeding db/db^+/−^ with either *mg53^−/−^* or tPA-MG53 mice. Through comparison of the six strains of mice (wild-type, *mg53^−/−^,* tPA-MG53, db/db*^−/−^*, db/db-mg53*^−/−^* and db/db-tPA-MG53), they demonstrated that diabetic mice strains (db/db, db/db-mg53^−/−^ and db/db-tPA-MG53) exhibited increased weight gain over 32 weeks, as compared to the non-db/db strains. The three DM strains demonstrated significantly elevated glucose levels through glucose tolerance testing at both 18 and 30 weeks of age, as compared to the three non-DM strains of mice [24]. Additionally, in comparing DM and non-DM mice, there were no differences in levels of MG53 (*p* = 0.113). While there is interdependence on skeletal muscle and muscle mass on insulin resistance, it does not appear that either whole-body ablation of MG53 (*mg53^−/−^* mice) or sustained constitutively overexpressed MG53 (tPA-MG53) has an impact on glucose handling and DM.

## 8. Conclusions and Prospects

Many studies have identified MG53 as an essential character of cell membrane repair which could be applied to treat injuries for multiple organs including myocardial infarction, muscular dystrophy, and damage in non-muscle organs including kidney, lung, skin and cornea. Muscle-derived MG53 can function as a myokine to facilitate repair of injury to remote organs. Sustained elevation of MG53 in circulation is safe in animal models, and does not impact glucose metabolism. While studies with rodents and large animal models support the therapeutic benefits of rhMG53 to treat acute injury to multiple organs, it remains to be established how protection against acute tissue injury can be translated to improvement in long-term function of the targeted organs. Moreover, studies targeting rhMG53 to treat chronic organ dysfunction or aging-related organ failure represent a potential area of future translational research. The recent establishment of an anti-inflammation function of MG53, in addition to its tissue repair function, opens a further area of therapeutic development of rhMG53 to treat both acute and chronic tissue injuries that are associated with inflammation.

## Figures and Tables

**Figure 1 cells-10-00122-f001:**
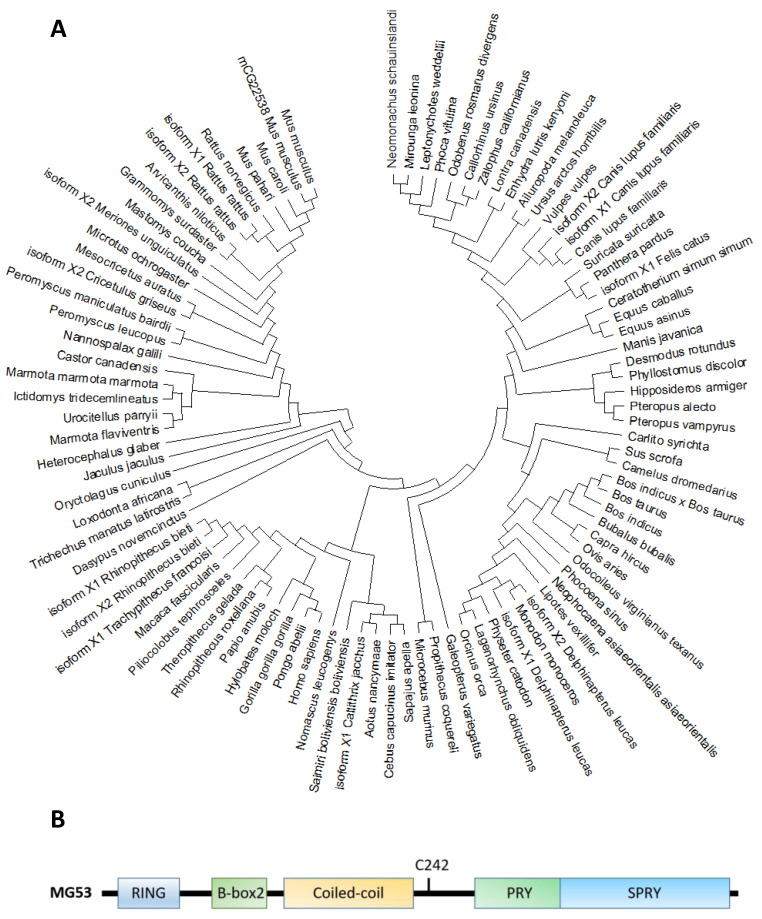
(**A**). Molecular phylogenetic analysis of MG53 proteins in different species by maximum likelihood. The evolutionary tree is presented to compare MG53 in different species. The maximum likelihood method and JTT matrix-based model were applied for inferring the evolutionary history [34]. The tree with the highest log likelihood (−7643.24) is shown. Neighbor-join and BioNJ algorithms were applied to a matrix of pairwise distances estimated using the JTT model, and the topology with superior log likelihood value was selected, then the initial tree(s) for the heuristic search were obtained. This analysis involved 95 amino acid sequences. There were a total of 599 positions in the final dataset. Evolutionary analyses were conducted in MEGA X [35]. (**B**). The primary amino acid sequence of MG53 contains the following characteristic tri-partite motifs (TRIM): RING, B-Box and Coiled-coil motifs in the amino terminus, and the PRY and SPRY domains at the carboxyl terminus. C242 is a critical cysteine residue involves in redox-dependent oligomerization of MG53 associated with cell membrane repair.

**Figure 2 cells-10-00122-f002:**
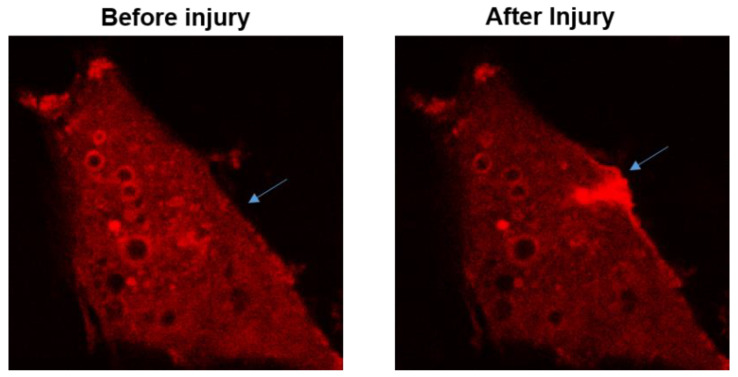
RFP-MG53 expressed in C2C12 cells moved quickly to the acute injury site following mechanical poking of the cell. Image shown on the right was taken at 60 s after injury. For more details, see Ref. [3].

**Table 1 cells-10-00122-t001:** Phenotype of *mg53^−/−^* mice and therapeutic application of rhMG53 protein.

Tissues	Phenotype of *mg53^−/−^* Mice	Therapeutic Application of rhMG53 Protein
SkeletalMuscle	muscle repair defect [3] reduced exercise capability [3,8]	muscle injury associated with exercise [3,9], muscular dystrophy [10]
Heart	defective cardiomyocyte repair [4,11] increased vulnerability to myocardial infarction	acute myocardial injury [12] chronic heart failure
Lung	defective alveolar structure and exacerbated lung injury under stress conditions [6]	acute lung injury, sepsis, chronic obstructive pulmonary disease [6,13,14]
Kidney	proximal tubular pathology [5] increased susceptibility to acute kidney injury	acute kidney injury [5] nephrotoxicity [15], chronic kidney disease
Cornea	reduced re-epithelialization and increased fibrosis following alkaline injury [7]	cornea injury and ulceration [7] prevention of cornea fibrosis

**Table 2 cells-10-00122-t002:** Schematic representation of MG53 conserved domain.

MG53 Species	Gene ID	Amino Acid (aa)	Identities (%)	Identities (%)	Protein Structure
RING	B-Box	SPRY
**Homo Sapiens**	NP_001008275.2	477	100	100	100	100	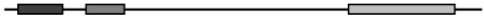
**Pan Troglodytes**	XP_001157628.2	477	100	100	100	100	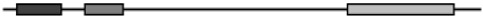
**Macaca Mulatta** **(Isoform 2)**	XP_001112866.1	477	98.95	100	100	97.93	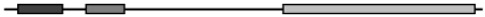
**Canis Lupus Familiaris (Isoform X2)**	XP_005621293.1	477	93.50	95.74	100	89.12	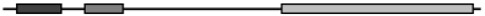
**Bos Taurus**	XP_002698119.1	482	94.71	95.74	100	90.16	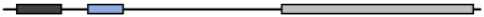
**Mus Musculus**	NP_001073401.1	477	91.19	95.74	97.56	87.56	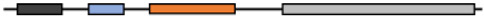
**Rattus Norvegicus**	NP_001071143.1	477	90.78	93.62	97.56	86.53	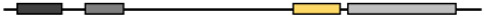
**Xenopus Tropicalis**	NP_001008188.1	477	59.14	53.19	72.50	62.63	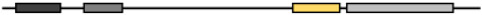

Notes: 
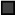
 RING finger 
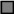
 B-box zinc finger 
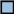
 B-Box-type zinc finger 
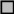
 SPRY domain 
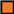
 Poly(hydroxyalcanoate) granule associated protein (phasin) 
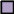
 SPRY-associated domain (PRY).

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
