# Peer review of "MG53, A Tissue Repair Protein with Broad Applications in Regenerative Medicine"

_cells, 2021, doi:10.3390/cells10010122_

Round 1
Reviewer 1 Report
The review article by Li et al. provides a convincing information on protein MG53 (TRIM72). Authors explain the importance of MG53 as a key component of cell membrane repair machinery in cells of different origin. Authors aptly highlighted the characteristics of TRIM72 mediated membrane repair and summarized the mechanisms. The therapeutic potential of recombinant human MG53 protein in repairing the cell membrane in multiple organ injuries is well explained. The article is well written and descriptive. Authors tried to put together all information very precisely; references are cited appropriately. There are no major flaws and authors did fairly good job. I recommend this for publication.
Authors can include a representative figure of MG53 corresponding to section 2.1 if it is within the scope of this manuscript (figure limitations).
Author Response
To reviewer 1
Response to Reviewers’ Critiques for MS #cells-1023075
Reviewer 1
The review article by Li et al. provides a convincing information on protein MG53 (TRIM72). Authors explain the importance of MG53 as a key component of cell membrane repair machinery in cells of different origin. Authors aptly highlighted the characteristics of TRIM72 mediated membrane repair and summarized the mechanisms. The therapeutic potential of recombinant human MG53 protein in repairing the cell membrane in multiple organ injuries is well explained. The article is well written and descriptive. Authors tried to put together all information very precisely; references are cited appropriately. There are no major flaws and authors did fairly good job. I recommend this for publication.
Authors can include a representative figure of MG53 corresponding to section 2.1 if it is within the scope of this manuscript (figure limitations).
Response: Thank you for your suggestion. We added Figure 1B to illustrate the domain structure of MG53.

Reviewer 2 Report
The review needs to be greatly improved in order to transfer an added value respect the state of art of the field
First of all, I’d like to ask the authors to better clarify the aims of the review by designing a coherent editing.
In addition, the authors should engaged to summarize the information available on the literature by giving a comprehensible overview and by avoiding to present the evidences collected to date simply as a list without any attempt that may contribute to a clear interpretation of the state of art.
I’d like to suggest to the authors to make more effort in synthetizing their message adopting explicative models that may take advantage of summarizing statements, figures and tables.
Finally, I strongly suggest to avoid any focus on inflammation (see introduction and conclusion sections) since the review does not have any organic paragraph dedicated to the role of MG53 on immune cells as well as on the modulation of innate and adaptative immunological responses mediating tissue regeneration.
Minor points
Abstracts
Please improve this section by better
- Focusing in premise As PLASMA membrane repair is related to regenerative tissue mechanism
- explaining the aim of the review
Introduction
- Please take attention in saying that “The functional communication is carried out by plasma membrane promoted cell-cell interactions via diverse cellular adhesions”. This is not true
- Clarify why the plasma membrane repairing mechanism may improve/innovate the regenerative medicine approaches
- Please avoid to use regeneration and repair as synonymous since it is not the case
From line 38 to line 46 there are several sentence explaining the scientific evidences collected to date without any link each other
From line 48 to line 51 there is just a list of tissues. Please avoid to do that
Line 52-53 I’d like to suggest to end the introduction by leaving the aims of the review more that the conclusions
General comments
A new editing may improve value of the review
A paragraph on the molecular characterization and cellular mechanism will be appreciate with a figure summarizing them
Considering that the manuscript is a review trying to address the focus on general mechanism in physiology and pathophysiology more than the role of MG53 in single district.
Try to summarizes the state of art of M53 underlying regenerative mechanism in explicative figure or tables. For samples: a table summarizing the effect of genetic ablation could be of help
The paragraph 2 should be dedicated exclusively to M53 distribution between tissue and species homologies
Please improve whole editing of the review
Delete paragraph 7 and insert this information on the paragraph dedicated to the MG53 mechanisms
Dedicated a separate paragraph to lung including the viral data
Conclusion
I highly suggest the Authors to avoid using this section to repeat statements already treated in the review instead to comment the potential clinical impact of the molecule, the R&I perspectives etc.
Author Response
To reviewer 2
Response to Reviewers’ Critiques for MS #cells-1023075
Reviewer 2
The review needs to be greatly improved in order to transfer an added value respect the state of art of the field. First of all, I’d like to ask the authors to better clarify the aims of the review by designing a coherent editing. In addition, the authors should engaged to summarize the information available on the literature by giving a comprehensible overview and by avoiding to present the evidences collected to date simply as a list without any attempt that may contribute to a clear interpretation of the state of art. I’d like to suggest to the authors to make more effort in synthetizing their message adopting explicative models that may take advantage of summarizing statements, figures and tables. Finally, I strongly suggest to avoid any focus on inflammation (see introduction and conclusion sections) since the review does not have any organic paragraph dedicated to the role of MG53 on immune cells as well as on the modulation of innate and adaptative immunological responses mediating tissue regeneration.
Response: We appreciate the reviewer’s constructive recommendations, and have followed her/his comments to make substantial changes in the Abstract, Introduction and Conclusion of this review article. As recommended by the reviewer, we deemphasize the anti-inflammation function of MG53 in the review article. Please see below for our detailed responses.
Minor points
Abstracts
Please improve this section by better
- Focusing in premise As PLASMA membrane repair is related to regenerative tissue mechanism
- explaining the aim of the review
Response: We have added a sentence to the Abstract: “Repair of injury to the cell membrane is an important aspect of physiology. Inadequate membrane repair function is implicated in the pathophysiology of many human disorders”. We also replaced “anti-inflammation” with “tissue regeneration” in the Abstract.
Introduction
- Please take attention in saying that “The functional communication is carried out by plasma membrane promoted cell-cell interactions via diverse cellular adhesions”. This is not true
Response: We agree, and have removed this sentence from the revised text.
- Clarify why the plasma membrane repairing mechanism may improve/innovate the regenerative medicine approaches
Response: We added the following sentences to the Introduction: “It is critical to maintain cellular integrity to ensure cell survival and tissue regeneration, as defects in this process are linked to the pathophysiology of many human diseases including muscular dystrophy, heart failure, lung injury and kidney disease”.
- Please avoid to use regeneration and repair as synonymous since it is not the case
Response: We agree with the reviewer, and have made changes in the text when appropriate.
From line 38 to line 46 there are several sentence explaining the scientific evidences collected to date without any link each other. From line 48 to line 51 there is just a list of tissues. Please avoid to do that. Line 52-53 I’d like to suggest to end the introduction by leaving the aims of the review more that the conclusions
Response: We thank the reviewer for these valuable suggestions. We have rewritten this paragraph in the revised version (line 36-48):
General comments
A new editing may improve value of the review.
Response: Yes, we have done extensive editing of this revised manuscript. We made tracking changes in the revised text.
A paragraph on the molecular characterization and cellular mechanism will be appreciate with a figure summarizing them. Considering that the manuscript is a review trying to address the focus on general mechanism in physiology and pathophysiology more than the role of MG53 in single district. Try to summarizes the state of art of M53 underlying regenerative mechanism in explicative figure or tables. For samples: a table summarizing the effect of genetic ablation could be of help
Response: Thank you for this valuable suggestion. We added a Table 1 to summarize the phenotype of mg53-/- mice and therapeutic application in treating injuries to different tissues.
The paragraph 2 should be dedicated exclusively to M53 distribution between tissue and species homologies
Response: Yes, we have section 2.1 focus on this topic.
Delete paragraph 7 and insert this information on the paragraph dedicated to the MG53 mechanisms
Response: We feel that paragraph 7 is an important topic to present to the readers, as we need to clarify the controversy for MG53 in modulation of insulin signaling and diabetes.
We also have expended section 5 to include an important topic “MG53-mediated repair on cornea protection” (section 5.4).
Dedicated a separate paragraph to lung including the viral data
Response: Thank you for this valuable suggestion. We have expanded section 6 on “A potential role of MG53 in treatment of acute lung injury”. This section has been largely re-written.
Conclusion
I highly suggest the Authors to avoid using this section to repeat statements already treated in the review instead to comment the potential clinical impact of the molecule, the R&I perspectives etc
Response: Yes, we revise this section to emphasize the clinical impact of MG53.

Round 2
Reviewer 2 Report
The Authors have greatly improved the manuscript even if I suggested to seek to make clearer the information to better contextualize the role of MG53.
To this aims:
Avoid to give statement without explanation and biological examples
i.eIntroduction section the authors have inserted this concept
It is critical to maintain cellular integrity to ensure cell survival and tissue regeneration, as defects in this process are linked to the pathophysiology of many human diseases including muscular dystrophy, heart failure, lung injury and kidney disease. But the main responses remain to be given yet: How with which mechanisms. A This is quite essential to understand and translated the potential role of this membrane pathways So I'd like to suggest the authors to summarize better these aspects and as suggested to made a figure to explain these aspect The table 1We added a Table 1 that summarize the phenotype of mg53-/- mice and therapeutic application in treating injuries to different tissues is useful even if for each lines the relative manuscript have to be inserted
but Table 1 aim is to define the role of MG53 not the mechanism involved as I suggested:
A paragraph on the molecular characterization and cellular mechanism will be appreciate with a figure summarizing them
Author Response
Please see the cover letter for our detailed response to reviewer 2's additional comments to our manuscript.
